# Effectiveness of Family-Based Behavior Change Interventions on Obesity-Related Behavior Change in Children: A Realist Synthesis

**DOI:** 10.3390/ijerph17114099

**Published:** 2020-06-08

**Authors:** Gemma Enright, Margaret Allman-Farinelli, Julie Redfern

**Affiliations:** 1Westmead Applied Research Centre, Faculty of Medicine and Health, University of Sydney, Sydney 2006, Australia; julie.redfern@sydney.edu.au; 2Department of General Practice, The University at Westmead Hospital, P.O. Box 154, Westmead 2145, Australia; 3Charles Perkins Centre, University of Sydney, Sydney 2006, Australia; margaret.allman-farinelli@sydney.edu.au

**Keywords:** childhood obesity, public health, behavior change, realist, family intervention

## Abstract

Effective treatment interventions for childhood obesity involve parents, are multicomponent and use behavior change strategies, but more information is needed on the mechanisms influencing behavioral outcomes and the type of parental involvement that is efficacious in behavioral treatment interventions with school-age children. This review aimed to understand key characteristics of programs that contribute to dietary and physical activity behavioral outcomes, and through which key mechanisms. This was a systematic review with narrative synthesis following PRISMA guidelines and realist analysis using RAMESES guidelines to explain outcome patterns and influence of parental involvement. Overall, the findings contribute to understanding the complex relationship between family barriers to behavior change, strategies employed in treatment interventions and behavioral outcomes. Implications for enhancing future policy and practice include involving parents in goal setting, motivational counselling, role modeling, and restructuring the physical environment to promote mutual empowerment of both parents and children, shared value and whole-family ownership in which intrinsic motivation and self-efficacy are implicit. These characteristics were associated with positive dietary and physical activity behavior change in children and may be useful considerations for the design and implementation of future theory-based treatment interventions to encourage habitual healthy diet and physical activity to reduce childhood obesity.

## 1. Introduction

Reducing the prevalence of childhood obesity is a global public health focus. In 2016, over 340 million school-aged children and adolescents aged 5–19 were overweight or obese [1]. Systematic reviews have found that being overweight as a child carries a high risk of being obese as an adult [2], and of bringing associated conditions such as cardiovascular disease and diabetes into earlier life [3]. Current and future health services are therefore crucial for preventing and managing childhood obesity [4], and behavioral strategies to support the adoption and improvement of health-related behaviors in children are becoming increasingly central to these programs [5].

Multicomponent interventions involving nutrition education, physical activity, behavior change and parenting strategies, targeting more than one energy balance-related behavior (physical activity, sedentary and dietary behaviors) are considered best practice in the literature [6,7]. Children are 10 times more at risk for obesity when both their parents are obese [8], and experts in childhood obesity recommend prevention and treatment of obesity in the primary school years should focus on parents, since this is when life habits are formed [9]. Whilst plenty of recent studies support evidence that parents exert high influence over energy balance behaviors in their children, making their involvement key to the success of child weight-management programs [10,11], findings have varied on the extent and for what ages parental involvement is successful [12]. Facilitating sustainable behavior change in children via public health interventions is challenging due to inhibiting influences of social and family barriers to healthy behaviors in participants’ wider lives [13]. A lack of parental or whole-family support can also impede new health behaviors being maintained long term [14]. Many authors have acknowledged the importance of parental involvement in childhood obesity treatment interventions and positive health-related outcomes. However, many ‘family-based’ interventions vary in their level and type of parent/carer involvement. Some interventions have focused on parents as the exclusive agents of change [15,16], where key elements consist of strengthening parents’ leadership skills, to take the focus off children being identified as the (obese) patient and overcome associated negative psychological issues and resistance to change. Other studies have promoted parent support of the child’s behavior change [16,17], with parents responsible for creating a supportive environment and empowering children to make informed choices. Finally, some studies have aimed to change both parent and child behaviors simultaneously [18,19], with children and parents making behavior change commitments together in the intervention.

A recent review of 70 RCTs (64 multicomponent, involving both parent and child) on the effects of diet, physical activity and behavior change interventions for overweight children [20] concluded that multicomponent interventions that incorporate diet, physical activity and behavior change strategies may be beneficial in achieving small short-term reductions in BMI, BMIz and weight in children aged 6–11 years. However, only a handful of the 56 studies that measured behavior change outcomes actually reported behavioral outcomes related to dietary (*n* = 2 studies) or physical activity (*n* = 8 studies) behavior change, and the review called for standardized reporting of key outcomes and mechanisms of behavior change. Implementation challenges associated with interventions based on behavioral theory present barriers to upscaling and translation into natural settings due to a lack of contextual information on factors that influence behavioral lifestyle outcomes [21,22,23]. In the past decade, systematic reviews of child and family-focused treatment interventions targeting obesity-related behavior change in overweight children have indicated small positive short-term outcomes on dietary and physical activity behaviors [5,24]. However, high heterogeneity across included studies (varied methodologies, settings, and intervention strategies) has been common to most systematic reviews, limiting the applicability of the findings and providing insufficient evidence for policy relevance or to support upscaling [5,25]. Overall, a better understanding is needed of patterns of outcomes across the family-based literature, and how the interaction between different program characteristics and behavior change strategies influence lifestyle behaviors such as heathy eating and physical activity in overweight and obese children.

The traditional systematic review methodology aims to minimize bias in order to provide a summary of the effectiveness of interventions in terms of their effect sizes. However, this approach can oversimplify the inherently contextual nature of interventions [26,27,28] and may not account for the complex relationship of interactions between context, mechanisms and outcomes [29]. As a result, realist reviews are emerging as an alternative ‘theory-driven’ approach to systematic reviews in public health research, as they are particularly relevant for understanding complex multicomponent interventions such as those prioritizing childhood obesity. However, there are currently no realist reviews on family-based behavior change interventions for overweight and obese children, measuring multiple obesity-related behavioral lifestyle outcomes and focusing on parental involvement.

This review aimed to enrich understanding of the ways child-focused behavior change treatment interventions contribute to reported obesity-related behavioral outcomes (specifically diet and physical activity behavioral outcomes as opposed to interventions based on healthy eating and physical activity behaviors for preventing obesity) and the influence of specific program characteristics and mechanisms.

The overall objective was to understand the key characteristics of programs that contribute to positive dietary and physical activity behavioral outcomes, and through which key mechanisms. More specifically, we aimed to explore;
What are the diet and physical activity behavioral outcomes of family-based behavioral treatment strategies?What are the key mechanisms by which family-based behavioral treatment strategies result in their outcomes? Specifically, how does theoretical grounding, targeted psychological variables, and behavior change techniques relate to one another and result in diet and/physical activity behavior change?What influence does the type of parental involvement have on the ways different mechanisms produce behavioral outcomes?

## 2. Materials and Methods

This research was a systematic review with narrative synthesis that followed PRISMA guidelines [30] and used realist analysis and RAMESES quality and publication guidelines [29]. Drawing from guidelines for realist reviews [27,28,29], an approach was developed based on six adapted iterative stages of realist reviews. The six stages are described below.

### 2.1. Literature Search

A comprehensive database search was conducted in March 2019 with an information specialist at University of Sydney Fisher Library in EMBASE (via Ovid), MEDLINE (via Ovid), and PsychINFO (via Ovid) using the following terms: obesity/overweight, behavior change/intervention/strategy/lifestyle, diet/eating/nutrition/food, physical activity/exercise, child/pediatric/youth/young person/family/school. The search strategy was iterative and involved adding, removing and refining search terms to retrieve search results with an appropriate balance of relevance and specificity. The search results were exported to EndNote X7 and deduplicated using the automatic deduplication feature and manual checking by G.E. The search strategies for each database are available on request. No language or date restrictions were applied, though studies published as an abstract only were excluded. In the event that a non-English paper was identified, it was reviewed against the inclusion criteria by people in our institution with the relevant languages as their native language to determine eligibility. Reference lists of other reviews were manually searched to find additional studies, and where possible, authors were contacted to obtain full text publications of relevant abstracts.

The review included both effective and non-effective interventions in diet and/or physical activity behavior change. Although BMI is commonly the primary outcome measured, this review focused on behavioral outcomes specifically in diet and physical activity, since the large body of literature on behavioral risk factors that have been associated with child obesity suggests that physical activity and eating habits are related, likely bidirectional in cause and effect, and should therefore be understood together. Furthermore, the literature has highlighted that BMI may be an inappropriate measure in children [31], and whilst BMIz factors in the age and gender of the child and may be a more optimal parameter research has indicated that the duration of weight-management programs is often too short for healthy weight and BMI reduction to manifest in growing children [32]. However, behavior changes over the same duration may set the child on a path for more gradual weight loss and help prevent the tracking of overweight and habitual clustering of obesity-related behaviors into adulthood [33,34]. Whilst prevention interventions have been indicated as critical in preschool years [35], we must not ignore the children who have already become obese as school-age children in order to prevent obesity and comorbidity tracking into adulthood and therefore this review focused on treatment interventions in school-age children.

Initial screening (*n* = 2037 articles) was conducted systematically by G.E. based on the title, abstract and keywords using the inclusion and exclusion criteria detailed in Table 1 to determine the a study was eligible for inclusion and likely to contain the relevant data.

### 2.2. Selection of Full Texts and Further Refinement

Full texts of relevant documents initially screened for inclusion were obtained (*n* = 88) and independently further refined for eligibility by two reviewers (G.E. and J.R). The further refinement was based on (i) ‘relevance’ in terms of each article’s contribution to addressing the research questions and contributing to theory building [27,29] and (ii) ‘rigor’, regarding the credibility and trustworthiness of methods used to generate the data via formal quality assessment. Articles were given ratings of ‘strong’, ‘moderate’ or ‘weak’ and disagreements about study eligibility were resolved through discussion between reviewers until consensus was achieved. Articles were categorized as having potentially strong contributions to answering the research questions if they contained detailed information regarding (i) involvement of the parent, (ii) underlying theory used in the strategy, discussion of psychological factors and/or mechanisms, and (iii) behavior change techniques (BCTs) with examples. Consideration was also given to studies focused on specific at-risk populations (e.g., low income), and studies that were particularly strong in one domain (e.g., extensive reporting of BCTs in relation to a theoretical underpinning). This process enabled reviewers to focus on data extraction and analysis of papers that provided a conceptually rich contribution while still including documents that were less conceptually rich for quantitative synthesis.

### 2.3. Extracting and Organizing Data—Systematic Review

Data extraction took place in two stages— ‘quantitative synthesis’ and ‘realist analysis’. In the quantitative synthesis, all included articles (*n* = 35) and their characteristic details were extracted into a detailed table in order to describe the studies. G.E. extracted the following details from each included study: author, year, study aim, country, design, associated program (if relevant), participants, setting, intervention (strategy, delivery, dose, duration of follow-up, and comparator if relevant), outcome measures relating to diet and physical activity behaviors (including other psycho-social behavioral variables), behavioral outcomes and unintended outcomes. A second author (J.R.) verified a random selection of extracted details. Data was then organized into a summary table of the study characteristics in relation to the proportion of studies demonstrating each characteristic and whether the study characteristic favored the intervention with positive dietary and/or physical activity outcomes. This table also summarized the proportion of studies based on a theoretical model, and the type of involvement of the parent.

### 2.4. Development of a ‘Program Theory’ Framework

An initial ‘program theory’ was developed as a framework for mapping selected context, mechanisms and outcomes for an iterative understanding of how intervention components from the studies included in the realist analysis have contributed to diet and physical activity behavioral outcomes. A program theory is *“an abstract description and/or diagram that lays out what a program (or family of programs or intervention) comprises and how it is expected to work”* [29] p. 24. A program theory can be a useful way of synthesizing existing evidence and hypothesized links between context, mechanisms and outcomes from multiple evaluations about how programs are understood to work, and where there are gaps in the evidence [36,37]. Our initial program theory was a template for the context of focus (‘C’); key barriers to family behavior change, mechanisms of focus (‘M’); parental involvement, behavior change techniques (BCTs) and targeted psychological, variables, and outcomes (‘O’). This set up potential pathways for combinations of these contexts and mechanisms that led to outcomes (Figure 1). As the review progressed, studies rated ‘strong’ in relevance (*n* = 11) were mapped onto the framework, causal pathways were identified and interlinked propositions explaining the causal relationships involved in how the interventions work were integrated into the program theory. Studies in the review rated as ‘moderate’ relevance (*n* = 15) were included in the discussions.

### 2.5. Extracting and Organizing Data—Realist Analysis

All documents with relevance ratings of ‘strong’ (*n* = 11) were scrutinized to identify key barriers as the rationale and premise for the strategy as the contextual focus (C). Documents were also scrutinized for details of mechanisms (M), focusing on (i) involvement of the parent (ii) behavior change techniques (BCTs) used in the intervention, and (iii) application of theory and psychological models in the intervention and interpretation of outcomes (e.g., targeted psychological variables to elicit behavior change). The Behavior Change Technique Taxonomy (BCTTv1) [38] was used to maximize replicability. G.E. undertook online training (http://www.bct-taxonomy.com) to ensure competence in the identification coding of BCTs. Verbatim sections of text were annotated and (color) coded manually by G.E. and identified themes and interpretations were independently reviewed by J.R. for consistency and refinement. Discussions between G.E. and J.R. were held to agree on the relevance of themes, and with the wider project team when there was a disagreement which could not be resolved. The full BCTTv1 labels with numbers were used in analysis and discussed under the 16 groupings description. Where available, information on full or partial Context-Mechanism-Outcome (C-M-O) configurations was extracted from individual studies. C-M-O configurations were then individually mapped on the program theory framework.

Analysis of the data entailed a combination of a deductive approach in which codes were based on the initial program theory framework, an inductive approach in which codes were identified from the documents reviewed, and a retroductive approach where inferences about underpinning mechanisms were made based on interpretations of the data contained within included studies [39]. A narrative synthesis approach [40] was selected as this style is recommended for use in reviews focusing on several questions that go beyond simply the effectiveness of an intervention.

### 2.6. Refining Program Theory

The final stage of the realist analysis was the refinement and validation of the program theory to ensure the final program theory made pragmatic sense. The review team revisited parts of the review that required re-scrutinizing and this process was continued until no new information was provided by the evidence, essentially reaching theoretical saturation [41]. Patterns in outcomes or ‘demi-regularities’ [27,29] across all study diagrams were configured into a final program theory, with the aim of identifying which context triggers which mechanisms and may lead to which outcomes.

### 2.7. Appraisal of Study Quality

The quality of each study was assessed with respect to its risk of bias according to the conventional approach, since the first part of the review was a systematic review based on pre-defined inclusion criteria. RCTs were assessed using the Cochrane Risk of Bias tool (random sequence generation and allocation concealment procedures, blinding of participants, personnel and outcome assessment attrition bias, and reporting bias) [42]. Non-randomized and observational studies were appraised against the Joanna Briggs Institute Critical Appraisal Checklist for Quasi-Experimental Studies [43]. Case reports, case series and qualitative studies were also evaluated using Joanna Briggs Institute Critical Appraisal Checklists. Studies were not excluded for realist analysis on the basis of the rigor assessment. G.E. assessed each study, and M.A-F. appraised 32% of the studies (*n* = 12/38 interventions) to ensure consistency across assessments. Studies were randomized using the function “randomize a list” in excel that generates a random number for each study ID. In addition, the ‘relevance’ of each study was assessed as per RAMESES guidelines with respect to its potential value to contribute to refining the program theory. Each study was given a relevance score of ‘strong’, ‘moderate’ or ‘weak’.

## 3. Results

### 3.1. Study Selection

From a total of 2037 abstracts screened for possible inclusion 88 full-text manuscripts were reviewed for eligibility, and thirty-five interventions (*n* = 36 studies) met final inclusion criteria [15,31,44,45,46,47,48,49,50,51,52,53,54,55,56,57,58,59,60,61,62,63,64,65,66,67,68,69,70,71,72,73,74,75,76,77] (Figure 2).

### 3.2. Risk of Bias Assessment

Methodological rigor varied within the different study designs (Appendix A). ‘Random sequence generation’ was the item for which most of the 18 RCT interventions (67%) were deemed to have the lowest level of bias (however, allocation concealment was inadequately addressed). This was followed by ‘selective reporting’ (50%), though 22% were deemed to have a high risk of bias in reporting (e.g., not using intent to treat analysis, varied ‘control’ arm comparisons such as comparing an intervention against usual treatment across different sites). Issues of participant and personnel blinding and blinding of outcome assessment had an unclear risk of bias due to insufficient information provided in the study; only 2/18 studies were deemed ‘low bias’ in participant/personnel blinding and only 1/18 study was deemed low bias for blinding of outcome assessment. Attrition bias was the item deemed as the highest risk of bias across the studies (44%) with high withdrawal and drop-out rates and low statistical power due to small sample sizes and missing data. Other biases identified in the studies included lack of generalizability due to specific populations being studied, and not being able to identify the relative contribution of variables; 83% of the studies were deemed to have an ‘unclear’ risk of other biases.

Regarding the quasi-experimental (pre–post) designs (*n* = 14 interventions), clarity on cause and effect was the biggest identified strength (86%), with selecting comparable groups and clear selection processes coming second (57%). However, not measuring outcomes in a reliable or clearly reliable way (79%), and incomplete or unclear follow-up (57%) were the biggest areas of weakness. Overall, only the case report studies (*n* = 2) and qualitative study were rated ‘strong’. Seven of the quasi-experimental studies were rated ‘moderate’ and seven were rated as ‘weak’ (Appendix A).

### 3.3. Study Characteristics

Table 2 provides a synthesis of included studies, grouped by characteristics and intervention effect. Table 3 summarizes the behavioral and psychosocial outcome measures, theoretical grounding and type of parental involvement across the 36 interventions included in the systematic review. Further details on each of the studies, including descriptions of the intervention strategies and the evaluation methods used to assess outcome effectiveness, are available upon request.

Pertaining to Table 2: Approximately one-third of the included interventions were evaluated using a randomized controlled trial (RCT) or cluster RCT (*n*  =  11, 31%), 15 (42%) were quasi-experimental comparison or observational studies, 7 (19%) were pilot or feasibility studies, and 3 (8%) were case reports or case series studies. Of the 36 studies, 21 (58%) demonstrated positive significant results on diet and/or physical activity behavior)—of which, 43% (*n* = 9) demonstrated significant positive results on both dietary and physical activity behaviors, 33% (*n* = 7) on only diet, and 24% on only physical activity behaviors (*n* = 5). The RCTs reflected a similar trend with more studies demonstrating positive between-group changes in only diet (27%) than only physical activity (18%). However, only 9% of RCTs demonstrated significant behavior changes in both diet and physical activity.

Most studies were published after 2010 (78%), highlighting the contemporary nature of family-based, childhood obesity work, and alluding to the growing momentum of research in this context. The majority of interventions were conducted in the USA, but other study locations were included. Almost half of the studies included less than 50 participants and nine (25%) prioritized children from families with low income (including Hispanic/Latino, African American, and Māori and Pacific Island populations). Of these priority population studies, five (56%) positively favored the interventions.

Most studies were conducted in a group-based setting (75%), and 56% of these demonstrated significant positive results on diet and/or physical activity behavior change, compared to only 25% of the exclusively individual (family) based interventions (*n* = 6/25). Conducting interventions in community settings appeared more effective on diet and/or physical activity behavior change than those that were predominantly school, or home based (41% compared to 17% and 0% respectively). 28% (*n* = 8) of the interventions included a remote delivery component, such as online platforms, telephone, telemedicine, and automated interactive voice response). Of these studies, 63% favored the interventions.

Intervention duration ranged from ≤1 four weeks to 12 months. Approximately 11% of studies included short-term post-intervention follow-up (≤6 months, *n*  =  4), with 14% presenting results of a long-term follow-up (≥12 months, *n*  =  5). Overall, of the 47% of studies that included a follow-up period, 29% (*n* = 5/17) demonstrated significant sustained dietary and/or physical activity behavior changes. The majority (53%) did not report on a post-intervention follow-up.

Interventions were delivered by a variety of facilitators; community registered dietitians/exercise physiologists, medical or health care staff, other clinicians (such as behaviorists, child psychologists, pediatric endocrinologists), local community experts on nutrition and exercise (such as chefs, bilingual volunteers, health educators, school teachers), and the research team. Interventions delivered by community registered dietitians or exercise physiologists were the most effective (73%). Physical activity was assessed using subjective methods (questionnaires, recall diaries and interviews) in 31 (86%) studies, whilst objective assessment (pedometry, accelerometry and direct observation) featured in 13 studies (36%). Evaluations based on self-reported physical activity measures were marginally more likely to report a positive intervention effect on diet and/or physical activity behavior (58%, compared to 46% of those using objective measures either alone or in conjunction with subjective measures). Only two studies used objective measures (direct observation) to assess dietary behavior.

Pertaining to Table 3: Just over half of the interventions (56%) targeted and measured only diet and physical activity behaviors, compared to 44% targeting and measuring other behaviors including; self-esteem, physical activity enjoyment, health related quality of life, self-efficacy, intrinsic motivation, and parental understanding of behavioral principles (Table 2). The interventions targeting only diet and physical activity were more successful in producing significant behavior change in diet and/or physical activity compared to the interventions targeting and measuring other behaviors (80% to 31% respectively).

Just under half of studies (44%) cited some theoretical grounding for their intervention. Theories underpinning the interventions were extremely varied, with 11 different theoretical models mentioned across the studies. Theory-based interventions appeared to be marginally more effective than those not referencing a particular theory (63% of those citing some theoretical background, and 55% of those citing none, were effective). Education was provided in almost all interventions; other frequently applied intervention strategies included goal setting and reinforcement of positive health behaviors (such as feedback, reviewing goals, praising, contracting, and social support, counselling, behavior substitution, environment control and role modelling). The use of theory, strategies and behavior change techniques used in the intervention is discussed in detail in the realist analysis.

Finally, whilst a key criterion for this review was parental involvement in the intervention, the studies divided into three different strategies involving the role of parents (Table 3). In 14% (*n* = 5) of studies, parents were identified as exclusive agents for change, 31% (*n* = 11) promoted parental support of the child’s behavior change, and 17% (*n* = 6) aimed to change both parent and child behavior simultaneously. All three strategies demonstrated similar proportions of significant effects on dietary and/or physical activity behavior change, with those promoting parent support of the child’s behavior showing the most success on diet and/or physical activity behavior change and particularly on both diet and physical activity behavior change together (45%).

### 3.4. Realist Analysis

#### Study Quality

Although all 35 studies (36 interventions) were potentially eligible for inclusion in the realist analysis, 11 provided sufficient information to describe outcome patterns (‘strong’ relevance) and hence contributed to the realist analysis [15,44,45,46,47,48,49,50,51,52,53]. Three of the studies were RCTs [45,48,53], seven were non-randomized quasi designs [15,44,46,47,49,51,52], and one was a pilot study which was included due its ‘strong’ relevance rating [50] (see Appendix A). Of these, all of the studies demonstrated some favorable changes in diet and/or physical activity behaviors; four studies demonstrated significant effects on both diet and physical activity (though not all effects were between groups) [44,46,52,53], three studies showed significant changes in dietary behaviors [15,45,47], and the remaining four studies found positive but non-significant effects (including qualitative findings) [48,49,50,51]. The studies displayed considerable variation in the context within which interventions were conducted, the intervention strategies implemented, and the intervention outcomes.

Despite this heterogeneity, the realist analysis enabled the extraction of demi-regularities (outcome patterns). These patterns are demonstrated in the specific configurations of interest grouped by type of parental involvement (discussed below). The final program theory was only able to summarize the most prevalent BCTs, psychological factors and mechanisms, since we were unable to establish clear links between specific BCTs, mechanisms and outcomes (Figure 3). This highlighted the complexity and multifaceted nature of treatment interventions for childhood obesity and the multitude of barriers to behavior change as starting premises for the interventions.

A wide variety of theories underpinned the studies included in the realist analysis, with most of the studies (*n* = 9/11) dedicated to one particular model. Social Cognitive Theory (SCT) [78] and Behavioral Economic Theory (BET) [79] were used by two of the studies, and one study used them together to underpin the intervention [48]. Only one study [39,47] used a combination of theories—Health Belief Model (HBM) [80] and Competence Motivation Theory (CMT) [81]. The BCTs most commonly used across the 11 studies included in the realist analysis were 1.1 ‘Goal setting (behavior)’ (*n* = 5), 1.2. ‘Problem solving’ (*n* = 5), 1.3 ‘Goal setting (outcome)’ (*n* = 5), 3.3. ‘Social support’ (*n* = 7), 6.1 ‘Demonstration of the behavior’ (*n* = 6), and 12.1 ‘Restructuring the physical environment’ (*n* = 6). Regardless of the combination of strategies and BCTs, intrinsic motivation (*n* = 5) and self-efficacy (*n* = 5) were identified as core psychological mechanisms for encouraging behavior change, with parent and child empowerment, ownership and shared value emerging as key mechanisms for behavior change. Matching intervention components to individual family’s readiness for change was also highlighted as important. Appendix A summarizes BCTs used in the 11 studies included in the realist analysis coded in accordance with the BCTTv1 [38], with examples from the study descriptions as a reference point. Specific configurations of interest are discussed below grouped by the type of parental involvement.

## 4. Parents as Exclusive Agents for Change

Interventions focused on parents as exclusive agents for change, aimed to address key barriers associated with controlling parenting styles, which can lead to low self-esteem, hopelessness and inability to self-regulate in the child [15]. Two studies contributed a combination of 1. ‘goals and planning’, 6. ‘comparison of behavior’ and 12. ‘antecedents’ behavior change strategies (Appendix A) which were effective in changing dietary behavior, through the mechanisms of increased self-efficacy and self-esteem, [15], and autonomy and intrinsic motivation [39] (Figure 4).

The two studies were based on behavior change theories, suggesting that small daily changes, goal setting, problem solving, and role modelling can influence behavior change (HBM), that children’s motivation (for physical activity specifically) is influenced by physical competence, social support and enjoyment (CMT), and recognizing the need for change, action planning, self-monitoring and feedback/reward is important in the goal setting process (SCT) [76,82,83]. SCT also posits that the intrinsic value of the goal influences successful goal setting and behavior change [84].

The influence of goal setting and action planning on behavior change was highlighted by Fisher et al. (2018), where parents were taught to set proxy ‘SMART’ (specific, measurable, achievable, realistic and time-framed) goals for their children (1.1 ‘Goal setting (behavior)’ and 1.4 ‘Action planning’). This approach drew specifically on the evidence in the literature suggesting that action planning can enhance goal setting (SCT). Fisher et al. (2018) [47] found that overall parents were significantly better at setting physical activity-related goals, such as reducing sedentary behaviors, than diet-related goals, but these goals did not translate into improvements in physical activity behavior in the children. Instead, significant changes were seen in dietary behaviors. The authors acknowledged that in the wider literature studies (in adults) comparing the nature of goal setting, there was greater success in goal setting related to autonomy (e.g., individually guided and goals self-determined rather than set collaboratively in a group) [85], which is supported by the fact that the intrinsic value of the goal is a significant predictor of goal setting and subsequent behavior change in adults [84]. In Fisher et al. (2018), goals were set by parents collaboratively as a group without any pre-assessment. This suggests that the intervention effects may have been strengthened (particularly for physical activity) if parents were allowed more autonomy in their goal setting, which may associate with increased self-efficacy, both in parents and subsequently children, if parents feel better able to model this behavior, as well as increased social support for the children, and an increase in enjoyment (intrinsic motivation). In Spurrier et al. (2016) [50], parents found it more difficult to model PA-related recommendations at home such as being active every day, having their children spend time outside, and ensuring family members were not eating in front of the television. It has been noted that parents may find it difficult to role model behavior such as decreasing screen time if they highly value screen time such as watching TV themselves [48]. Furthermore, it has been suggested in the wider literature that decreasing sedentary time (e.g., screen time) may be easier than increasing physical activity [86]. Fisher et al. (2018) also hypothesized that parents, when setting ‘proxy’ goals for their children, may be less likely to be successful when more than one area of behavior change is focused on at a time.

Golan et al. (1998) [15] taught parents to alter the family sedentary lifestyle, provide a prudent diet, and decrease the family’s exposure to food stimuli (12.1 ‘Restructuring the physical environment’ and 12.3 ‘Avoidance/reducing exposure to cues for the behavior’). The strategy aligns with the literature on role modelling (e.g., CMT), where the parents act as both model and mediator. By strengthening parents’ leadership skills through education and encouragement Golan et al. (1998) found the children ate less between meals because there were fewer temptations. Compared to the alternative approach of children being made responsible for their own behavior change, the authors believed parents being the exclusive agent for change reduced children’s’ pressure of being singled out as the (obese) patient and reduced their resistance to change, especially when able to mimic their parents modelling healthy behaviors. Golan et al. (1998) also found, however, that although most parents succeeded in ‘mediating’ changes in the family environment, they apparently failed to model the behaviors, as there was no significant reduction in parental weight after the intervention, and no correlation between change in parental eating behavior and children’s overweight reduction.

## 5. Promoting Parent Support of the Child’s Behavior Change

Four of the studies in the realist analysis used strategies based on assumptions that parents creating a supportive environment is more effective than prescriptive and controlling approaches [46,48,49,50]. Several systematic reviews in the wider literature have flagged parental involvement and targeting of the home environment as key aspects related to intervention success [87,88], and it has been documented that reinforcement increases activity more than restriction [89]. The interventions in the present review aimed to address barriers to behavior change associated with negative emotional consequences of controlling parenting styles and obesogenic environments created by overweight parents [46], struggles translating strategies to improve the home environment [50], and motivation related to the strong reinforcement of sedentary activities compared to active alternatives [46,48], (Figure 5).

These studies contributed a combination of a wide range of behavior change techniques including 1. ‘Goals and planning’, 2. ‘Feedback and monitoring’, 3. ‘Social support’, 4. ‘Shaping knowledge’, 6. ‘Comparison of behavior’, 8. ‘Repetition and substitution’, 10. ‘Reward and threat’, 11. ‘Regulation’, 12. ‘Antecedents’ and 15. ‘Self-belief’ (described in detail with examples from the documents in Suppl. 2). All four studies used a combination of at least four different behavior change techniques to effect behavior change and were effective in significantly changing both dietary and physical activity behavior through the mechanism of increased intrinsic motivation [46,48]. Mechanisms of self-efficacy [48,49] and autonomy and resilience [49] led to a significant (net) reduction in sedentary activity and moderate increase in high-intensity physical activity and sustained (anecdotal) longer-term positive trends in diet and physical activity behavior (9 months duration of effect beyond the intervention) [49].

Two key strategies were identified in the realist review: (i) reinforcing intrinsic motivation by replacing an undesired behavior with something more valued by both children and parents, and (ii) empowering both parents and children to recognize their choices by increasing children’s resilience and parent’s flexibility.

*(i)* *Shared value:* Reinforcing intrinsic motivation by replacing an undesired behavior with something more valued by both children and parents.

Strategies based on ‘reinforcement’ in the realist review were underpinned by BET, which focuses on understanding how children make choices about being sedentary or active and on reinforcing value. Epstein et al. (2004) incorporated this theory, basing their intervention on the assumption that children either increase physical activity (“substituting”) or reduce energy intake (“complimenting”) when sedentary behaviors are reduced. The authors demonstrated that stimulus control (e.g., 12.1 ‘Restructuring the physical environment’) or reinforcing reduced sedentary behaviors (e.g., 10.8 ‘Incentive (outcome)’; includes ‘Positive reinforcement’—a contract reinforcement system, where families selected appropriate reinforcers and assigned point values for the reinforcers to be earned for meeting behavior change goals), could produce significant decreases in sedentary behaviors. They found those who substituted active for sedentary behaviors showed a greater rate of change (in zBMI) at 6 months and 12 months than those who did not substitute [46]. Maddison et al. (2014) [48] adopted a similar approach to encourage children to reduce screen time in their home environment, drawing from BET and SCT. The authors offered alternatives to screen-based media (e.g., 8.2 ‘Behavior substitution’) and encouraged parents to implement strategies in the home to monitor and budget their child’s media usage (e.g., 2.1 Monitoring of behavior by others without feedback). Both studies also encouraged families to use praise and positive reinforcement (e.g., 10.4 ‘Social reward”). Whilst the budgeting device used in Maddison et al. (2014)’s intervention did not gain traction with participating families, there were some moderate positive trends across dietary and physical activity behaviors in response to behavior substitution. The authors offered two reasons for the non-significant result. Firstly, wider research has shown that children use screen-based media because it is highly valued and rewarding [90], so substituting with non-financial rewards, such as playing board games or increasing physical activity, may have been less valued by the children. Secondly, the intervention encouraged parents to act as role models for decreasing screen time, which may have proven difficult if parents also highly valued screen time.

*(ii)* *Empowerment:* Empowering parents and children to recognize their choices

Strategies based on empowering parents and children to own their choices were underpinned by resiliency psychosocial frameworks and ecological theories. These theories encourage interventions to promote resilience in the child and recognize the whole family and broader cultural context as being influential in creating negative through patterns and inflexibility when making new choices [91,92]. Further, SCT emphasizes supporting parents to make changes in weight-related cognitions and behaviors themselves so as to model their behaviors to their children.

Siwik et al. (2013) incorporated these theories by investigating a strategy whereby children committed to a “turn off challenge” and their parents committed to supporting them. This involved children and parents self-observing their own TV behavior (e.g., 2.3. ‘Self-monitoring of behavior’), analyzing “traps” and coming up with steps to circumvent them (e.g., 1.2 ‘Problem solving’). The intervention focus was on reducing negative and ’inaccurate’ thought patterns (e.g., 11.2 ‘Reduce negative emotions’; 15.4 ‘Self-talk’) by increasing resilience in the child and giving parents permission to become flexible and change their response to their child’s behavior around eating and activity. This was done through problem solving and motivational interviewing. This approach allowed participants to modify their lifestyle individually in relation to their goals, a process that fostered autonomy (key to intrinsic motivation), self-efficacy and self-esteem.

Siwik et al.’s strategy was congruent with the premise in the literature that a key parenting practice is to create a protective environment in the home applicable to the whole family to avoid stigmatizing obese children [93]. This supports Golan et al. 1998 [15] and Fisher et al. 2018 [47], who used the same philosophy regarding the parental role. The mechanisms through which this worked were similar across these three studies (self-efficacy, self-esteem and autonomy (Figure 4)) and collectively led to the empowerment of both children and parents to recognize and make informed choices. In the words of Siwik et al. (2013) [49], *“…the model empowered parents and children by providing a nonjudgmental framework to make lifelong changes. …acknowledged children as important agents for change and allowed them to make educated choices not driven by rebellion or conflict. It built on multiple previous works of Epstein* [94] *that children with the most control over their physical activity in the intervention were the most active years later and carried that thread forward about food and beverage choices… [and] even the most authoritarian parents began to relinquish unnecessary levels of control by week 8 as they witnessed their children making reasonable choices”*.

An emphasis on being non-judgmental, reducing the stigma of the “obese child”, and being supportive and sensitive to socioeconomic determinants of obesity [49,50] was a strong theme in the context of promoting parent support of the child’s behavior change. In Spurrier et al. (2016) particularly, who demonstrated only very moderate effects on diet and physical activity behavior within individual families (most related to diet such as children’s access to food in the home and lunchbox, greater access to vegetables and restriction of high fat/salty foods), the impact of ‘chaotic’ socioeconomic factors on the ability of families to instigate change became more obvious as the study progressed. These observations were facilitated by visits being undertaken ‘in situ’ in the family home. Parents found it especially difficult to model behaviors such as not eating in front of the TV and taking children to parks. The authors suggested that the lack of intervention effect may reflect the need for more intensive and ongoing support for social and emotional issues of both child and family, to prevent these complex socioeconomic factors becoming a priority for families over positive lifestyle change. It is also worth noting that socioecological theory, on which the intervention was based, suggests that macro-environments exert greater influence on obesity than parent’s ability to optimize the family home environment [95], indicating that interventions might take a more integrated home/school collaborative approach such as ‘Conjoint Behavioral Consultation’ [65] (Table 2).

## 6. Aimed to Change Both Parent and Child Behaviors Simultaneously

Three studies included in the realist review used strategies based on the assumption that positive parent health-related outcomes plus involvement in the intervention are key tools to achieving sustainable change for their children [44,45,51] (Figure 6). SCT posits that behavior interacts in an ongoing reciprocal manner with personal cognitions and the surrounding environment, including (specifically in children), the cognitions and behaviors of parents. Therefore, for overweight children to make healthy physical activity and dietary changes, changes may also be required in their parents’ weight-related cognitions and behaviors. The literature is sparse, however, on the relationship between program outcomes for parent participants and the long-term sustainability of healthier weights for their children.

Two key barriers to behavior change these studies sought to address were (i) individual differences in rates of learning and parenting behaviors [45], and (ii) negative emotional issues such as overweight children suffering low self-esteem [51]. Closely related to individual rates of learning, recognizing children’s needs in the different stages of change was also acknowledged [52]. Strategies based on matching the rate of progress to the needs of individual participants at different stages of change were adopted, which aligned with the transtheoretical model of change (though this was not acknowledged by any of the authors). Epstein et al. (1994) (RCT) matched the introduction of skills for participants to ‘master’ and reinforcement for behavior change to participant’s skill level and progress [45], and Wong and Cheng (2013) employed motivational interviewing with a range of other BCTs to address different stages of change [52]. Note that Wong and Cheng (2013) designed their intervention with parental influence in mind rather than aiming to change and measure the behavior of parents as well as children. The study was included in this discussion due to its focus on the stages of change.

Overall, both studies demonstrated significant behavioral changes to diet [45] or to physical activity [52] through the mechanisms of self-efficacy [45,52] and intrinsic motivation [52]. Whilst both interventions utilized multiple BCTs from 1. Goals and planning (both using 1.5 ‘Review behavior (goal(s)’), the BCT strategies overall looked quite different. Wong and Cheng (2013) emphasized goals and planning, framing/reframing (13.2 in BCCTv1) and self-talk (15.4 in BCCTv1), and Epstein et al. (1994) emphasized praise and reinforcement (10.4 ‘Social Reward’; 10.8 ‘Incentive (outcome)’ and 12.1 ‘Restructuring the physical environment’; 12.3 ‘Avoidance/reducing exposure to cues for the behavior’). Despite there being no obvious comparable pattern in BCTs employed to address stages of change, both studies were underpinned by principles of Social Learning Theory (SLT) [96]. SLT emphasizes the acquisition of skills within a social group, through observation, imitation, and modelling, and promotes the development of an accurate perception of self and acceptance of others, which Wong and Cheng (2013) addressed in their intervention in order to encourage readiness to change (e.g., 13.2 ‘Framing/reframing’; 15.4 ‘Self-talk’).

Regarding addressing barriers relating to negative emotional issues associated with being an obese child, Watson et al. (2015) [51] employed a wide range of BCTs in the ‘GOALS’ program, focusing on 1. ‘Goals and planning’ (1.1 ‘Goal setting (behavior)’, 1.2 ‘Problem solving’ (x3 examples), 1.3 ‘Goal setting (outcome)’, 1.4 ‘Action planning’, and 1.8 ‘Behavioral contract’). The authors additionally used; 2. ‘Feedback and monitoring’ (x3 BCTs), 5. ‘Natural consequences’ (x3 BCTs), 6. ‘Comparison of behavior’ (x3 BCTs), and 8. ‘Repetition and substitution’ (x3 BCTs). This mix of BCTs contributed only to positive trends in diet and physical activity behavior changes through the (targeted and measured) mechanism of increased self-esteem. Anderson et al. (2015) [44], in contrast, reported only two BCTs (also used by Watson et al. (2015) [51]); 3.1 ‘Social support (unspecified)’ and 6.1 ‘Demonstration of the behavior’ and demonstrated significant changes to diet and physical activity via directly targeting an increase in self-efficacy and intrinsic motivation.

In GOALS [51], parents reported very moderate changes in self-esteem at 6 months (social acceptance), which were sustained at 12 months. Whilst this may have contributed longer term to children’s willingness to participate in physical activity outside the GOALS context, the authors suggested that the focus on ‘weight loss’ rather than ‘healthy behaviors’ in the program may have inhibited the effects of the intervention strategies. It was interesting in this study that at 12 months follow-up, the children who had lost the most weight during the intervention had the most improved self-perceptions. However, it was difficult to decipher whether weight loss lead to improved self-esteem or whether improved self-esteem lead to weight loss. Regardless, it is important that obesity treatment interventions help parents/carers to promote a healthy body image in children by focusing on healthy behaviors rather than weight [51]. In the ‘Taking Steps Together’ program [44], parents were encouraged to role model for their children and work towards behavior change as a team—thereby, creating whole-family ownership of the program and self-efficacy. These were important aspects of the intervention that supported the BCTs employed; families created their own guidelines about attendance, participation, and graduation, and through structured activities and dialogue children and their parents identified the barriers and resources for healthy living in their communities. Building participant ownership of the program and self-efficacy may empower families to identify their strengths and challenges and feel in control to set their own pathway to better health. This aligned with Siwik et al. (2013) [49] in that increasing a sense of control and self-efficacy in children (and parents) can influence sustained improved health choices. The program also reinforced the message that eating healthy foods and being physically active must be enjoyable, which fosters intrinsic motivation. This was underpinned by self-determination theory (SDT) [97], which posits that the most powerful and sustainable form of motivation is intrinsic motivation, where individuals pursue specific behavior because it provides personal satisfaction and enjoyment. Therefore, activities in the intervention were ‘experiential’; exercise was ‘fun’ and healthy foods were ‘delicious’ [44].

In terms of whether significant changes by parents accounted for children’s results, the findings were mixed. Epstein et al. (1994) [45] demonstrated a significant improvement in parental knowledge of behavioral principles but was unable to determine whether this was influenced by mastery criteria or contingent reinforcement (i.e., which treatment group), and overall the authors were unable to determine which strategy was more effective for diet and physical activity behavior change. In Anderson et al. (2015), though the child participants (from low-income and ethnically diverse backgrounds) showed significant changes in dietary and physical activity behaviors from baseline to the end of the 16 week program, their parents showed broader improvement in health-related behaviors. Watson et al. (2015) [51] demonstrated only small changes in child self-esteem (social acceptance) at 6 months, but this was maintained at 12 months, where no declines were reported by parents in the family’s diet and physical activity behavior or on the child’s self-confidence. Interestingly, improvements to parents own physical activity levels differed to those of their children with parents focusing on structured exercise and children on sports participation, suggesting that the modelling of behavior by parents was influential, whilst the children had enough self-confidence to make their own choices and incorporate their own interests.

## 7. Discussion

This review aimed to determine the key mechanisms for the ways in which family-based behavioral treatment interventions for overweight and obesity in community settings contribute to diet and physical activity behavioral outcomes. Overall, more than half of the 36 identified family-based interventions promoting diet and physical activity were effective in positively changing behaviors. In terms of key mechanisms by which family-based behavioral treatment strategies resulted in their outcomes, there was considerable heterogeneity and the final program theory was not able to establish clear links between the most prominent BCTs, psychological factors and outcomes. This highlighted the complexity and multifaceted nature of treatment interventions for childhood obesity and the multitude of barriers to behavior change as starting premises for the intervention components.

This review found no clear set of ‘ingredients’ for an ideal program but importantly certain program features were common across successful dietary and/or physical activity behavior outcomes. These features were promoting intrinsic motivation and self-efficacy through empowering children and their parents, fostering shared value between children and parents, and whole-family ownership. Social Cognitive Theory was a prominent theoretical foundation in this review. The use of such theories of motivation has been common in child obesity interventions aiming to influence engagement in positive health behaviors by promoting self-efficacy and intrinsic motivation [11,98]. There were no obvious patterns of BCTs pertaining to specific barriers addressed by the interventions, or by type of involvement by parent. However, the five BCTs used most frequently were goal setting, problem solving, social support, demonstration of the behavior, and restructuring the physical environment. This reflects the literature, which has also found combinations of these strategies to be frequently used (and successful) in influencing behavior change in childhood obesity treatment interventions [99,100,101]. This also highlights the numerous possible ways of promoting intrinsic motivation, self-efficacy and desired family values key to positive behavior change in children.

This review reinforces the impact and importance of parental involvement on mechanisms associated with behavioral outcomes. Although there was no standout types of parental involvement, promoting parent support of the child’s behavior demonstrated a slightly higher proportion of diet and/or physical activity behavior change. Cohesion and agreement within families was the key to successful behavior change, which echoes the wider literature [102]. This review highlighted some further key considerations for future practice involving parents. Firstly, a focus on autonomy, particularly in goal planning with parental involvement, may help to translate goal planning into improved behavior change in children (particularly in physical activity), as evidenced in this review [39]. Autonomy is one of the constructs of intrinsic motivation and related to successful goal setting and behavior change [84,91]. Literature is growing in its ability to demonstrate the importance of autonomous motivation for physical activity and healthy eating behavior change in overweight and obese children [103,104]. Secondly, counselling approaches such as motivational interviewing involving the whole family may be key to supporting parents support their children. This review identified that counselling in the family context can help present alternatives to restrictive/controlling parenting, promote resilience, address readiness to change, and prevent social and emotional barriers to behavior change (such as low self-esteem) from dominating over healthy behavior change in the family [49,50]. Motivational interviewing in particular has been shown by the wider literature to increase positive dietary habits and physical activity [105,106]. Third, parent-only interventions may help address children’s’ self-esteem issues and reluctance to attend weight-management programs. The review highlighted that this is contingent on fostering mutual empowerment, value and ownership [61,64]. This is congruent with the wider literature, which emphasizes the importance of baseline parenting style and parental self-efficacy for mediating improvements in children’s health-related behaviors [107,108].

This review and realist analysis offers important opportunities for future research. For example, the mechanisms and clinical implementation of the novel approach to parental involvement in child obesity treatments referred to as ‘Conjoint Behavioral Consultation’ offers potential [65]. Further, the pivotal role of the family physician and their potential person-centered counselling capacity for both parents and children is important for future research to examine the influence of the family physician on family behavior change. More detailed analysis and research into strategies suited to specific cultures and subpopulations would be beneficial. It is quite possible that specific priority populations as well as rural populations have unique barriers [109]. One such barrier may include accessibility. Davis et al. (2013) [58] has conducted a line of work investigating the delivery of treatment interventions to families via telemedicine and found this delivery method to be as effective as the telephone and in-person physician visits. It was beyond the current review to fully analyze the context-mechanism-outcome pathways for specific populations, but important for future research to focus on (particularly in the current Covid-19 pandemic, when many people are desiring minimal group and face-to-face contact to avoid the community spread of disease).

This review has several strengths including the use of a standardized reporting format (namely the taxonomy for identification, reporting and assessment) helping to minimize variations and aid information synthesis in future reviews. A number of weaknesses must be acknowledged as well. Firstly, the cause of childhood obesity is not at all simple and we did not consider factors such as genetic predisposition, sociodemographic factors, and psychological and environmental factors. A social ecological approach would be ideal for the focus of further reviews to acknowledge social and environmental influences on behavior change. However, the realist analysis did not draw upon stakeholder expertise as modelled by a recent realist analysis [110] which involved community and international stakeholders throughout the process in order to validate and refine the program theory, to ensure it was of practical relevance. There was also great variation in study designs, and without all studies having a control group, we are limited in being able to assert whether healthy changes made by participants were directly related to the intervention or to other confounding variables. Ideally, further analysis would look at the findings of the randomized trials included in this review separately to the non-randomized trials to determine whether they differ. Whilst our inclusion criteria specified that children must be aged between 7 and 13 years, the age range in some of the studies was much broader. This must be acknowledged as a limitation of the review given that the impact of parental strategies and mechanisms of change are likely to differ with the age of the child. The principles of building self-efficacy and intrinsic motivation as mechanisms for healthy behavior change were important elements identified in the realist analysis. However, they were not directly measured in most of the studies. It was also difficult to assess the longer-term effects of the included interventions because of missing data, or a lack of reporting of specific diet and physical activity behaviors. Finally, whilst behavioral outcomes were chosen for the focus of this review, it would be important for future research to investigate why some interventions were effective in changing behaviors and BMI/BMIz whilst others changed behaviors but not BMI/BMIz, and clarify whether this is related to the length and intensity of the intervention program.

## 8. Conclusions

Implications of this research include involving parents in goal setting, motivational counselling, role modeling, and restructuring the physical environment to promote mutual empowerment of both parents and children, shared value and whole-family ownership in which intrinsic motivation and self-efficacy are implicit. These characteristics were associated with positive dietary and physical activity behavior change in children and may be useful considerations for the design and implementation of future theory-based treatment interventions to encourage habitual healthy diet and physical activity to reduce childhood obesity.

## Figures and Tables

**Figure 1 ijerph-17-04099-f001:**
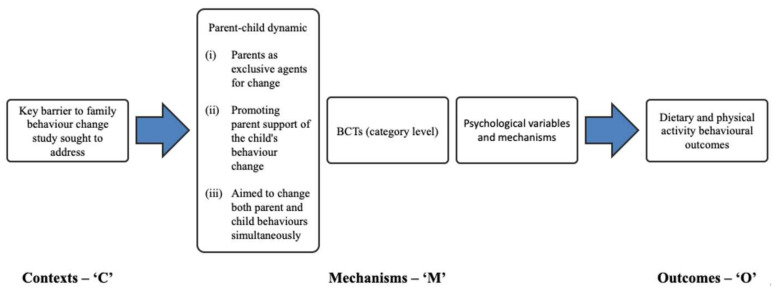
Program theory framework.

**Figure 2 ijerph-17-04099-f002:**
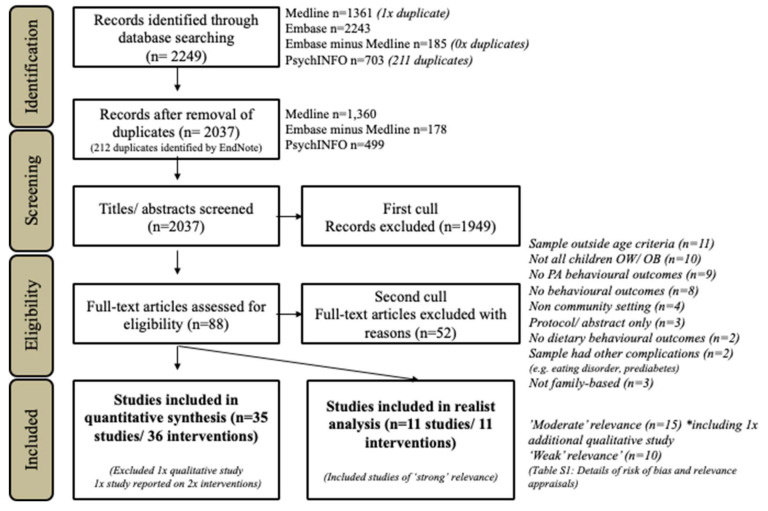
Study selection PRISMA flow diagram.

**Figure 3 ijerph-17-04099-f003:**
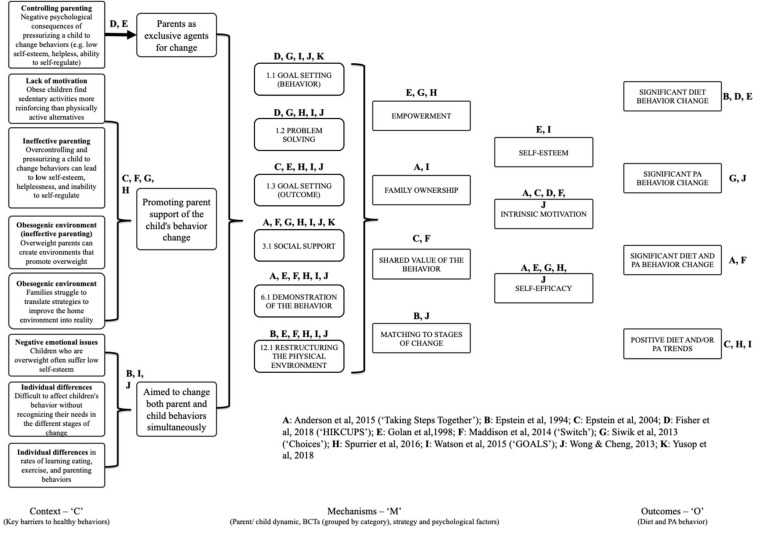
Final program theory. A: [44] (‘Taking Steps Together’); B: [45]; C: 46; D: [47] (‘HIKCUPS’); E: [15]; F: [48] (‘Switch’); G: [49] (‘Choices’); H: [50]; I: [51] (‘GOALS’); J: [52]; K: [53].

**Figure 4 ijerph-17-04099-f004:**
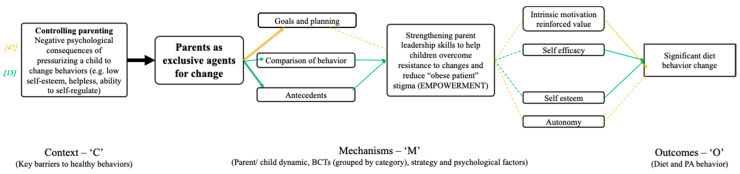
C-M-O pattern based on interventions with parents as exclusive agents for change. Solid arrows indicate configuration between context, mechanisms and outcomes evidenced by the included studies, and dashed arrows depict those configurations that were hypothesized but not evidenced.

**Figure 5 ijerph-17-04099-f005:**
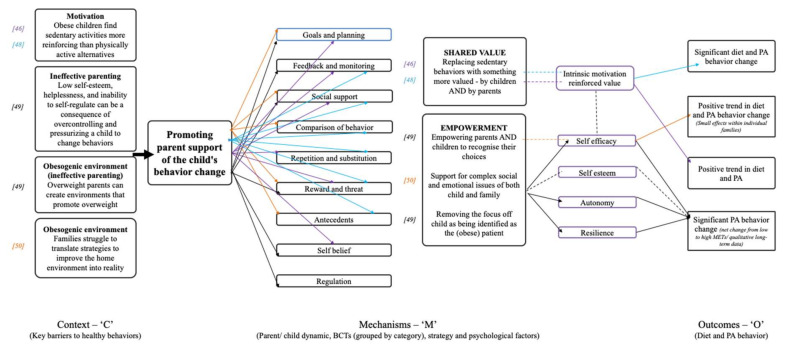
C-M-O pattern based on interventions promoting parent support of the child’s behavior change. Solid arrows indicate configuration between context, mechanisms and outcomes evidenced by the included studies, and dashed arrows depict those configurations that were hypothesized but not evidenced.

**Figure 6 ijerph-17-04099-f006:**
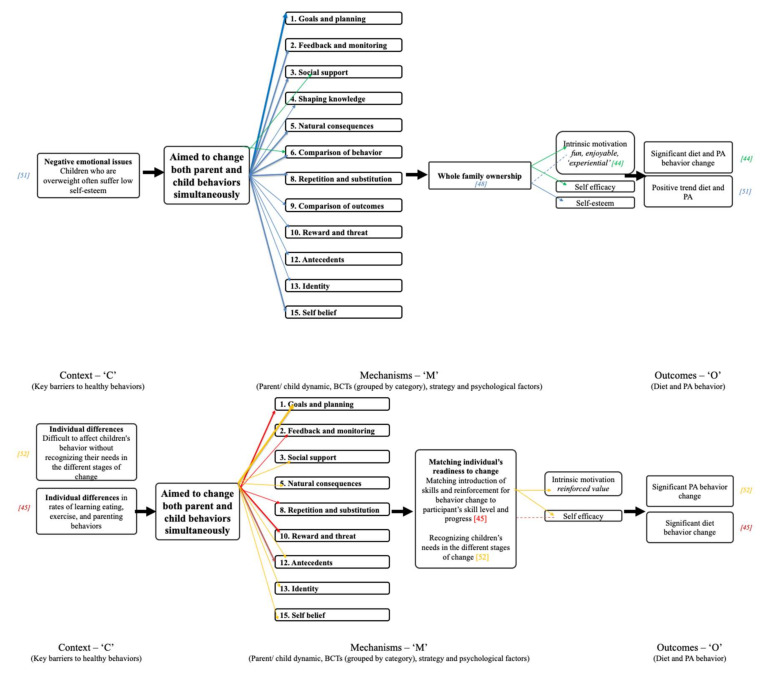
C-M-O patterns based on interventions aiming to change both parent and child behaviors simultaneously. Solid arrows indicate configuration between context, mechanisms and outcomes evidenced by the included studies, and dashed arrows depict those configurations that were hypothesized but not evidenced.

**Table 1 ijerph-17-04099-t001:** Inclusion and exclusion criteria.

Study Characteristic	Inclusion Criteria
Country and language	Any country or language.
Status	Published peer-reviewed and grey literature.
Dates	No date restriction.
Design	RCTs, cluster RCTs, all quasi-experimental and observational designs (including cohort, cross sectional, longitudinal, case control studies), case series and case reports, and qualitative studies.
Setting	All community-, school-, and home-based interventions, including initiatives trialed within existing weight-management programs. Remote interventions (i.e., delivered via telephone or other technology) were also included. Articles were excluded if based on interventions carried out in hospital or research-based institutions.
Intervention duration and follow-up	Any intervention or follow-up duration.
Population	Children aged 7–13 years classified as overweight or obese, (including but not restricted to the definition of Centre’s for Diseases Control and Prevention, where body mass index (BMI) must be greater than the 85th percentile for age and sex). Articles were excluded if the primary focus was adolescent or adult obesity.
Intervention	Child obesity-focused, family-based interventions. All eligible articles were ‘multicomponent’ and included (i) education, (ii) behavior change strategies as either stand-alone or multicomponent intervention, and (iii) parental involvement.
Outcomes	*Primary (essential):* Behavioral outcomes relating to both dietary and physical activity behavior change, including subjective or objective measures of increases or decreases in healthy eating or an increase or decrease in physical activity or sedentary behavior. *Secondary (not essential)*: Social cognitive variables (e.g., self-esteem, physical activity enjoyment, self-efficacy, intrinsic motivation) and qualitative outcomes such as barriers and facilitators to implementing new behaviors.

**Table 2 ijerph-17-04099-t002:** Characteristics of 36 interventions included in systematic review of family-based weight-management programs with diet and physical activity behavioral outcomes.

Study Characteristic	Proportion % *(n = 36 Interventions)*	% Favored Intervention (Diet and PA)	% Favored Intervention (Diet)	% Favored Intervention (PA)
**Study Design**				
RCTs/cRCTs	31 (11)	9 (1)	27 (3)	18 (2)
Quasi-experimental *(non-randomized)/*observational studies	42 (15)	27 (4)	27 (4)	7 (1)
Case reports/case series	8 (3)	67 (2)		
Pilot/feasibility studies	19 (7)	29 (2)		29 (2)
Qualitative component	8 (3)	33 (1)		
**Year of Publication**				
2018–2019	8 (3)		67 (2)	33 (1)
2016–2017	14 (5)		20 (1)	
2010–2015	56 (20)	35 (7)	10 (2)	15 (3)
1994–2009	22 (8)	25 (2)	25 (2)	13 (1)
**Study Location**				
USA	47 (17)	35 (6)	12 (2)	6 (1)
UK	6 (2)			
Europe (excl. UK)	8 (3)		33 (1)	
Australia	11 (4)	25 (1)	25 (1)	
Other locations*	(28 (10)	20 (2)	30 (3)	40 (4)
**Sample Size**				
<50	42 (15)	27 (4)	7 (1)	20 (3)
51–200	53 (19)	26 (5)	26 (5)	11 (2)
≥200	6 (2)		50 (1)	
Priority Population *(low socioeconomic status)*				
>50% Hispanic/Latino	11 (4)	50 (2)	25 (1)	
Other priority populations**	14 (5)	20 (1)		20 (1)
**Intervention Duration**				
≤1 month (up to 4 weeks)	3 (1)	100 (1)		
1–2 months (up to 8 weeks)	6 (2)	50 (1)		
>2–3 months (up to 12 weeks)	28 (10	10 (1)	20 (2)	20 (2)
>3 months (12 weeks+)	58 (21)	29 (6)	19 (4)	14 (3)
≥12 months	6 (2)			
**Follow-up (Post-Intervention) Periods Reported** **Sustained effects shown**				
Short term: up to 6 months	11 (4)	50 (2)		25 (1)
Medium-term: 7–12 months	22 (8)	13 (1)		
Long term: >12 months	14 (5)		20 (1)	
No follow-up	53 (19)	na	na	na
**Intervention Deliverer*****				
Community registered dietitians/exercise physiologists	42 (15)	27 (4)	27 (4)	20 (3)
Medical/health care staff	25 (9)	11 (1)	22 (2)	22 (2)
Other clinicians	33 (12)	25 (3)	8 (1)	
Local community experts on nutrition and exercise	36 (13)	15 (2)	15 (2)	23 (3)
Research team/research assistant	11 (4)	50 (2)		
Automated component	3 (1)			100 (1)
Unspecified	14 (5)	20 (1)	40 (2)	
**Group or Individual Intervention**				
Predominantly group-based sessions	15 (27)	19 (5)	26 (7)	11 (3)
Exclusively individual	25 (9)	44 (4)		22 (2)
Setting				
Predominantly school	14 (5)	40 (2)	20 (1)	40 (2)
Predominantly community center	58 (21)	19 (4)	29 (6)	10 (2)
Predominantly home	8 (3)			
Collaborative	6 (2)	50 (1)		50 (1)
Remote delivery component	22 (8)	50 (4)		50 (1)
**PA Outcome Measure** **✢**				
Subjective	86 (31)	26 (8)	16 (5)	16 (5)
Objective	36 (13)	31 (4)	15 (2)	
**Diet Outcome Measure** **✢**				
Subjective	100 (36)	25 (9)	19 (7)	14 (5)
Objective	6 (2)	50 (1)		
General notes— Some studies demonstrating significant effects were uncontrolled, so it is not possible to ascertain whether effects were a direct result of every intervention (e.g., [44]) Not all significant effects demonstrated were significant between groups (i.e., a study design that included two interventions that both showed significant within-group effects from baseline to end of treatment (e.g., [46]) All positive significant effects have been reported in this review since there was no restriction on study design, unless the study was a RCT and specifically reported that there were no between-group differences (e.g., [48,60]) Studies with only significant diet OR PA may also have shown positive trends in the non-significant behavior (e.g., [62]) Where a study demonstrated significant between-group effects in one behavior (e.g., PA) but only within-group effects for the other behavior (e.g., diet), the between-group result will be reported in this review (e.g., [52,53,74])				

PA: physical activity. RCT/cRCT: randomized controlled trial/cluster randomized controlled trial. * Thailand, Canada, Mexico, Israel, China, NZ, Malaysia, and Hong Kong. ** >50% Māori (Indigenous) and Pacific children, deprived areas of Liverpool, UK, and low-income African Americans. *** Some studies employed more than one deliverer (excludes control group deliverers). ✢ Some studies employed both subjective and objective methods to measure diet and PA behavior.

**Table 3 ijerph-17-04099-t003:** Outcome measures, theoretical grounding, and type of parental involvement in 36 interventions included in systematic review of family-based weight-management programs with diet and physical activity behavioral outcomes.

Study Characteristic	Proportion % *(n = 36 Interventions)*	% Favored Intervention (Diet and PA)	% Favored Intervention (Diet)	% Favored Intervention (PA)
**Outcome Measures**				
Diet- and PA-related only	56 (20)	30 (6)	30 (6)	20 (4)
Additional behavioral/psychosocial variables*	44 (16)	19 (3)	6 (1)	6 (1)
**Theoretical Grounding (in relation to BCTs/mechanisms)**				
Theory identified	44 (16)	25 (4)	19 (3)	19 (3)
No theory identified	56 (20)	25 (5)	20 (4)	10 (2)
**Type of Parental Involvement**				
Exclusive agents for change	14 (5)		40 (2)	
Promoting support of child’s behavior change	31 (11)	45 (5)		9 (1)
Aimed to change both parent and child behavior	17 (6)	17 (1)	17 (1)	17 (1)
Parent acknowledged as influential**	39 (14)	21 (3)	29 (4)	21 (3)

PA: physical activity. BCTs: behavior change techniques. *Additional behavioral/psychosocial measures included; physical activity enjoyment/self-esteem, e.g., Harter Self-Perception Profile; health-related quality of life; emotional issues, e.g., Child Behavior Checklist (CBCL); feeding variables, e.g., Behavioral Pediatrics Feeding Assessment Scale (BPFAS); Intrinsic motivation, e.g., The Intrinsic Motivation Inventory for Weight Management (IMI-WM)); self-efficacy; parental understanding of behavioral principles, e.g., Knowledge of Behavioral Principles Questionnaire ‘KBPQ’); the home environment, e.g., The Home Environment Inventory. ** Parental involvement did not fit any of the other categories.

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
