# Peer review of "Effectiveness of Family-Based Behavior Change Interventions on Obesity-Related Behavior Change in Children: A Realist Synthesis"

_ijerph, 2020, doi:10.3390/ijerph17114099_

Round 1

Reviewer 1 Report

For inclusion criteria: any country or language I am wondering how you can handle this. I did not find any other languages for the literaure. According to my knowlege, there are a few publication in this field in Chinese and other language as well.

Author Response

Effectiveness of family-based behavior change interventions on obesity-related behavior change in children: a realist synthesis

REVIEWER 1

  1. For inclusion criteria: any country or language I am wondering how you can handle this. I did not find any other languages for the literature. According to my knowledge, there are a few publication in this field in Chinese and other language as well.

Response: The search identified 4 non-English papers (Dutch - Mulkens et al, 2007; Spanish - Diaz et al, 2015 + Bustos et al, 2015; Korean - Seo et al, 2005) and all were excluded as per below. In order to screen these, we arranged review according to the inclusion criteria by people in our institution with the relevant languages as their native language. We have now clarified this process in the revised manuscript (page 4 paragraph 1, lines 151-153).

Reviewer 2 Report

Thank you for a well constructed paper with very relevant research questions regarding behavioural outcomes, mechanisms of action, BCTs and contextual factors pertinent to family-based behavioural treatment interventions for children aged 7-13y with overweight/obesity.

The methodology used for the systematic review and realist analysis shows application of rigour and best practice guidelines.  Presenting results as C-M-O patterns was particularly informative given the focus on exploring casual linkages.

Your discussion makes some interesting points in relation to future intervention research.  Your conclusion summarises your research findings succinctly, however, considering the need for child obesity treatment interventions, it may be more powerful to present your concluding remarks in terms of implications for research, practice and policy.  

Author Response

REVIEWER 2

  1. Your discussion makes some interesting points in relation to future intervention research.  Your conclusion summarises your research findings succinctly, however, considering the need for child obesity treatment interventions, it may be more powerful to present your concluding remarks in terms of implications for research, practice and policy.  

Response: In the revised manuscript we have revised the statement in the conclusion as suggested (page 25, paragraph 2, line 944-959)

Reviewer 3 Report

Dear Authors of the paper "Effectiveness of family-based behavior change interventions on obesity-related behavior change in children: a realist synthesis",

I want to thank for the possibility to read and review this comprehensive work. It is easy to imagine how much time and effort that lies behind this submission. 

Even though I have some conserns that I will try to make more spesific.

Introduction.

You describe that "Children are 10 times more at risk for obesity when both their parents are obese [8], and experts in childhood obesity recommend prevention and treatment of obesity in the primary school years should focus on
parents, since this is when life habits are formed. I realize that this paper is focusing on interventions, but miss one or two sentences about the impact of nature, biology and  heritage when this focus is outlined. 

Method

"This research was a systematic review with narrative synthesis that followed PRISMA guidelines
[27] and used realist analysis and RAMESES quality and publication guidelines [26]. Drawing from guidelines for realist reviews".

All aspects of the methods used in relation to review of the relevant literature are very well described. My knowlendge about doing and reporting a realist synthesis is limited. Even though I find the initial descriptions easy to follow. When it comes to reporting the results and discussion of them this paper becomes to complicated for me. 

Results

Table 2 is confusing. It seems like you try to report about everything and then it becomes very difficult to follow. It is not easy to understand what is reported in each comumn, and parents involvment in different interventions are reported at last. This table should be spilt in two (as a minimun) - so that  demographic information about each study were displayed in one table and results related to the research questions in another table. 

In the research questions you talk about causality and mechanisms  - I cant' see how this is reported in between all other information. It is to much tekst about too many aspects.

Discussion/conclusion

 I hoped you would sum up the results, answering your research questions in an understandable way, but again - this becomes to complicated. 

In the conclusion chapter you list some factors that is important to address when designing interventions. I am sorry to say that I was not able to see how you cam eto this conclusions

Author Response

REVIEWER 3

  1. You describe that "Children are 10 times more at risk for obesity when both their parents are obese [8], and experts in childhood obesity recommend prevention and treatment of obesity in the primary school years should focus on parents, since this is when life habits are formed. I realize that this paper is focusing on interventions, but miss one or two sentences about the impact of nature, biology and  heritage when this focus is outlined. 

Response: Thank you for this suggestion, in the revised manuscript we have added several sentences that recognise the impact of nature, biology and heritage (page 2, paragraph 2, lines 57-60).

  1. Table 2 is confusing. It seems like you try to report about everything and then it becomes very difficult to follow. It is not easy to understand what is reported in each column, and parents involvement in different interventions are reported at last. This table should be spilt in two (as a minimun) - so that  demographic information about each study were displayed in one table and results related to the research questions in another table.

Response: We agree with this suggestion and have split the table in two in the revised manuscript (Table 2a and Table 2b – pages 9-12). We have also added clarifying sentences in the Results section before each table to help break down the information, and have given more detail on the characteristics of the studies included in the realist review (page 12/13, paragraph 2/1, lines 397-404).  Based on this feedback we have also taken the opportunity to simplify and delete text in Table 1 where possible.

  1. In the research questions you talk about causality and mechanisms - I can’t see how this is reported in between all other information. It is too much text about too many aspects.

Response: We have revised and simplified this section of the revised manuscript (page 3, paragraph 3, lines 112-133) and adjusted the abstract to reflect these changes also.

  1. Discussion/conclusion - I hoped you would sum up the results, answering your research questions in an understandable way, but again - this becomes to complicated. 

Response: We have taken on this feedback and have considerably revised the Discussion and Conclusion, so they are more understandable and focused directly on the research aims (pages 25-29, lines 757-959).

  1. In the conclusion chapter you list some factors that is important to address when designing interventions. I am sorry to say that I was not able to see how you cam eto this conclusions

Response: A similar issue was raised by reviewer 2 and we have revised the conclusion to make it more straightforward with practical suggestions (page 25, paragraph 2, lines 944-959)

Reviewer 4 Report

The authors present a review designed to identify and synthesize the contexts within which interventions with successful behavioral change operate, understand the key characteristics of programs that contribute to positive dietary and physical activity behavioral outcomes, and through which key mechanisms using a realist analysis approach. The topic area is important and the realist approach is a novel and interesting way to look at this. There are, however, significant areas of concern that would question the usefulness and possibly validity of these findings. Some specific concerns and suggestions are outlined below.

Overall the paper is overwhelming to read and digest and could benefit from greater focus. For example, focusing only on the studies where they can conduct and report on the realist analysis rather than presenting both the narrative review on the larger number of studies  and realist analysis on the smaller subset would be one way. 

The statement in the abstract that the most effective treatment interventions for childhood obesity involve parents, are multi component, and use behavior change strategies... is not entirely correct as success with involvement of parents is age dependent. This abstract also gives the impression that the authors will look at effective interventions which usually implies favorable changes in BMI. It would help to clarify this up front.

It would be important and interesting to researchers to better understand causal mechanisms and possible differences as to why some interventions were effective in changing behaviors and making favorable changes in BMI vs. those that changed behaviors but not BMI. Although healthy behaviors are clearly important, favorable changes in BMI are ultimately an important and common goal of most interventions with obese and overweight children. Alternatively do not restrict this to studies with overweight and obese children. Prevention studies to promote healthy behaviors have been done in general populations of children.

The inclusion of such a wide variety of study designs raises concerns about the validity and generalizability of the findings. Case series and pilot studies in particular would best be excluded. There is still concern for confounders in the non-randomized designs that could account for the behavior changes rather than the intervention - mainly the quasi experimental and observational- but one could look at these separately from the randomized trials to see if one had different findings. 

Finally , the age range of the studies possible appears to be quite wide and parental strategies and causal mechanisms would likely be quite different. A narrower focus on young children would be more relevant and useful in practice.  

Author Response

REVIEWER 4

  1. Overall the paper is overwhelming to read and digest and could benefit from greater focus. For example, focusing only on the studies where they can conduct and report on the realist analysis rather than presenting both the narrative review on the larger number of studies  and realist analysis on the smaller subset would be one way.

Response: Similar issues have been raised by Reviewer 3 and hence we have refined the manuscript for clarity and language, including the Results and Discussion. We have also simplified Table 1 and Table 2 (splitting it in two). We would prefer to keep all the inclusion criteria and structure of the review because we wanted it to be detailed and comprehensive. We believe it best to keep this as a single document at this stage and hope the revisions help with clarity. We have also simplified the aims of the review as well as the Discussion and Conclusion in line with other reviewer comments.

  1. The statement in the abstract that the most effective treatment interventions for childhood obesity involve parents, are multi component, and use behavior change strategies... is not entirely correct as success with involvement of parents is age dependent. This abstract also gives the impression that the authors will look at effective interventions which usually implies favorable changes in BMI. It would help to clarify this up front.

Response: We have refined the language in the abstract to clarify our aim to further understand the type of parental involvement that is efficacious in FBTs with school age children (page 1, paragraph 1, lines 15-17). We have also included a sentence in the introduction that reflects the variation of research findings on what level and for whom parental approaches should be used (page 2, paragraph 2, lines 57-60). We have included a sentence in the Methods explaining the focus on behavioural outcomes rather than BMI (page 4, paragraph 2, lines 156-170), and a sentence in the limitations explaining that a social ecological approach would be ideal for the focus of further reviews to acknowledge social and environmental influences on behaviour change (page 24, paragraph 2, lines 908-916).

  1. It would be important and interesting to researchers to better understand causal mechanisms and possible differences as to why some interventions were effective in changing behaviors and making favorable changes in BMI vs. those that changed behaviors but not BMI. Although healthy behaviors are clearly important, favorable changes in BMI are ultimately an important and common goal of most interventions with obese and overweight children. Alternatively do not restrict this to studies with overweight and obese children. Prevention studies to promote healthy behaviors have been done in general populations of children.

Response: We have revised the Discussion (limitations) to reflect the need to better understand why some interventions were effective in changing behaviours and BMI whilst others changed behaviours but not BMI (page 24, paragraph 2, lines 932-936). We also added several sentences to the Methods to clarify why we focused on behavioural outcomes rather than BMI, and why treatment interventions with overweight and obese children, rather than prevention initiatives (page 4, paragraph 1, lines 156-170).

  1. The inclusion of such a wide variety of study designs raises concerns about the validity and generalizability of the findings. Case series and pilot studies in particular would best be excluded. There is still concern for confounders in the non-randomized designs that could account for the behavior changes rather than the intervention - mainly the quasi experimental and observational- but one could look at these separately from the randomized trials to see if one had different findings.

Response: We agree that this is complicated, but we intentionally included a variety of study designs and discussed the quality and risk of bias in these studies separately to the randomised controlled trials. However, we have expanded on the limitations to acknowledge that confounding variables could be a concern and risk of bias in the interpretation of behaviour change in the nonrandomised studies included in the realist analysis (page 24, paragraph 2, lines 920-924).

  1. Finally, the age range of the studies possible appears to be quite wide and parental strategies and causal mechanisms would likely be quite different. A narrower focus on young children would be more relevant and useful in practice.

Response: The inclusion criteria was children aged between 7-13 years. However, some of the studies did have a much broader age range. In this instance we included studies where at least 50% of a broad age range were within 7-13 years. However, we have noted this as a limitation in the Discussion section of the revised manuscript (page 24, paragraph 2, lines 924-928)

Round 2

Reviewer 4 Report

The authors have been responsive to the review. There are no further comments or concerns on the revised manuscript. 

Author Response

The authors have been responsive to the review. There are no further comments or concerns on the revised manuscript. 

Response: Dear reviewer 4. Many thanks for reviewing the revised manuscript. We have read through again thoroughly and conducted a further spell check.